# Relationship between Religiosity and Subjective Well-Being among Middle-Aged Korean Women: Focused on Roles of Existential Consciousness and Savoring Beliefs

**Eunjin Beak, Sung-Jin Chung and Kyung-Hyun Suh ***

Department of Counseling Psychology, Sahmyook University, Seoul 01795, Korea; 1000eunjin@hanmail.net (E.B.); sjchung@syu.ac.kr (S.-J.C.)
* Correspondence: khsuh@syu.ac.kr

**Abstract:** This study identified the relationship between religiosity and the subjective well-being of middle-aged Korean women, and examined a double mediating effect model of existential consciousness and savoring beliefs on this relationship. The participants of this study were 285 middle-aged Korean women, aged between 40–60 years. The PROCESS Macro 3.5 Model 6 was used to analyze the sequential double mediating effects. The results revealed that existential consciousness and savoring beliefs were positively correlated with the subjective well-being of middle-aged women, whereas their intrinsic religiosity was not significantly correlated with subjective well-being because it was positively correlated with negative emotions, as well as with life satisfaction and positive emotions. In a sequential double mediation model for the subjective well-being of middle-aged women, the direct effect of intrinsic religiosity on savoring beliefs was negatively significant after adjusting for indirect effects through existential consciousness. The sequential indirect effect of existential consciousness and savoring beliefs on the intrinsic religiosity and subjective well-being of middle-aged women was significant. Without these indirect effects, intrinsic religiosity negatively influenced middle-aged women's subjective well-being in this model. These results suggest that existential consciousness plays an important role in the subjective well-being of middle-aged women.

**Keywords:** religiosity; subjective well-being; existential consciousness; savoring beliefs; middle-aged women

## 1. Introduction

### 1.1. Religiosity and Well-Being

Frankl (1967) emphasized that religiosity plays an important role in adapting to human life. It is not yet clear how religion positively influences individual adaptation or well-being; however, several researchers have argued that religion and religiosity have positive effects on individual life (Batson et al. 1993; Donahue 1985; Pargament 1977). Chatters et al. (2008) found that people who are religious or regularly attend religious services are mentally healthier than those who do not. Earlier studies have provided evidences that religiosity relieves anxiety and depression (Lea 1982; Peterson and Roy 1985). These findings suggest that religiosity has a positive effect on quality of life.

Religiosity, which can be defined as a religious orientation with a commitment to religious beliefs and activities, may improve an individual's psychological well-being (Hathaway and Pargament 1990); however, this is not its only effect because there can also be psychopathological elements in religiosity (Rosenstiel and Keefe 1983). This may be related to "the double-edged sword of religion (Ben-Nun Bloom and Arikan 2012). In Bergin's meta-analysis, studies on the relationship between religiosity and mental health appeared in two opposite ways (Bergin 1983). In his study, this relationship was divided into adaptive and maladaptive groups among people with high religiosity. In one quarter of the 24 studies, religiosity was negatively correlated with mental health, whereas it was

positively correlated with mental health in half of the studies. The effect size of religiosity on mental health in these 24 studies ranged from −0.32 to +0.82. Bergin's study clearly showed that there are factors of religion that positively affect mental health as well as factors that negatively affect mental health.

Religiosity may have elements that can positively influence subjective well-being (Villani et al. 2019). However, it seems that this influence is less than that of mental health. The results of Witter et al.'s meta-analysis of the relationship between religiosity and subjective well-being indicated that the correlation coefficient between religiosity and subjective well-being in 56 studies ranged from 0.14 to 0.25 (Witter et al. 1984), meaning that religiosity accounted for only between 2% and 6% of subjective well-being. The lower degree of influence of religiosity on subjective well-being in previous studies may be due to measurement-related limitations or the complexity of these two concepts (Koenig 2012). Some argue that specific types of religiosities are related to subjective well-being, whereas others are not. For example, Argyle and Hills (2000) believed that only the pursuit of transcendence or spirituality—that is, intrinsic religiosity not related to social interests— was significantly correlated with people's subjective well-being. This may be because there are variables that mediate the relationship between religion and subjective well-being or mental health, such as spiritual experiences, image of God, optimism, self-esteem, religious coping, social support, etc. (Cheadle and Dunkel Schetter 2018; Fabricatore et al. 2004; Nooney and Woodrum 2002; Testoni et al. 2016; Wnuk 2021). However, we assumed that factors within one's religiosity or specific recognition created by religiosity may improve one's subjective well-being.

### 1.2. Roles of Existential Consciousness and Savoring Beliefs in Person's Life

We chose existential consciousness—that is, the sense of existence referring to a present state of recognizing the meaning, purpose, and fullness in one's life (Ownsworth and Nash 2015)—as such a variable. It is assumed that subjective well-being can be improved when religiosity induces existential consciousness. Frankl, who emphasized the importance of religiosity, also illuminated the value of existentialism included in religiosity (Frankl 1967). It is common sense that religion should meet the needs of believers for a sense of existence (Royce 1962). Therefore, existentialism is a major topic in theology (Jansen 1966). The existential function of intrinsic religiosity can reduce anxiety related to the afterlife and provide psychological tolerance (Van Tongeren et al. 2013). Because discovering meaning in life can act as an important factor in well-being (Steger 2018), we believe that existential consciousness may play an important role in people's subjective well-being. In addition, Wnuk and Marcinkowski (2014) found that existential variables, such as hope and the meaning of life, mediate the relationship between religious or spiritual functions and psychological well-being. Thus, this study assumes that existential consciousness mediates the relationship between religiosity and subjective well-being.

People's existential senses, such as mindfulness, can help them savor life and allow them to experience joy and happiness (Robbins 2021). Thus, we assumed that savoring beliefs created by existential consciousness increases subjective well-being. Bryant et al. (2005) conceptualized savoring beliefs as thoughts that enhance the appreciation of positive experiences and emotions through reminiscence. They consist of (1) anticipation, that is, the enjoyment of expected positive events; (2) being in the moment, that is, expanding the present positive experience; and (3) reminiscing, that is, recalling memories to re-experience and savor positive emotions again (Bryant et al. 2005). Bryant et al. (2005) suggested that savoring beliefs inevitably increase subjective well-being, and a study empirically found that they can increase people's feelings of happiness (Jose et al. 2012). Because a recent study found that a state of elevated existential consciousness, such as a flow of consciousness, can create an optimal state in which to enjoy life (Burt and Gonzalez 2021), this study assumed that existential consciousness may influence people's subjective well-being through savoring beliefs.

### 1.3. Subjective Well-Being of Middle-Aged Korean Women

This study focused on subjective well-being as represented by overall life satisfaction or happiness rather than by psychological well-being, which emphasizes value, meaning, and functional aspects (Diener 1984; Ryan and Deci 2001). Traditionally, subjective well-being comprises frequent positive emotions, infrequent negative emotions, high life satisfaction, and high levels of subjective happiness (Ryan and Deci 2001). Life satisfaction is distinguished from emotional appraisal because it is more cognitively driven than emotionally driven (Diener et al. 1985). Moreover, emotions reflect the frequency or the degree to which they are experienced; thus, the level of subjective well-being is high if people experience fewer negative emotions, experience more positive emotions, and are more satisfied with their lives.

Because middle-aged women in Korea experience various changes in the transition period of life, and it is important to maintain their quality of life due to the complex manifestation of problems during that period (Lee et al. 2014), this study was conducted with middle-aged women. In addition, middle-aged women in Korea experience extensive life-related stress, as well as fear of loss and loneliness, because they are bound by their families as wives, mothers, housewives, and daughters-in-law (Seo and Jeong 2020). There are also reasons why women are more religious than men (de Vaus and McAllister 1987), but the relationship between religiosity and the well-being of middle-aged women has long been of interest to researchers (Genia and Cooke 1998). It was found that middle-aged women's sense of existence, such as discovering the meaning of life, was positively correlated with their well-being (Cho and Jeong 2017). Therefore, this study explored the religiosity and well-being of middle-aged Korean women.

### 1.4. Hypothesis of This Study

In this study, we hypothesized that religiosity directly increases the subjective well-being of middle-aged women. The religiosity of middle-aged people is positively, directly and indirectly, to existential consciousness, related to subjective well-being. It is possible that middle-aged women also experience more savoring beliefs when they have higher religiosity; therefore, this study attempted to verify a model that includes a path from religiosity to subjective well-being through savoring beliefs. The model also includes a sequential double mediation effect of existential consciousness and savoring beliefs between religiosity and the subjective well-being of middle-aged women (Figure 1).

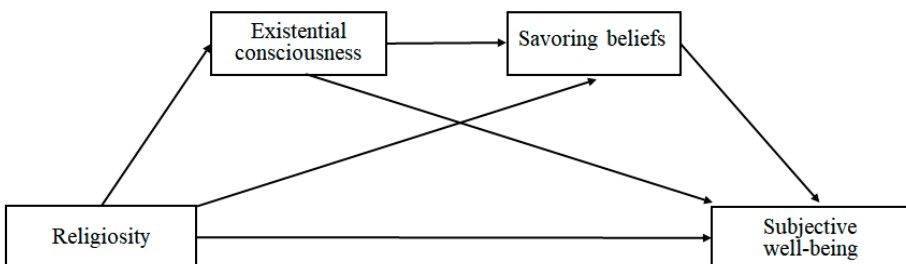

**Figure 1.** Double sequential mediation model.

## 2. Methods

### 2.1. Participants

A total of 285 middle-aged Korean women, aged 40–60 years, were selected using convenience sampling. Data were collected using a Google questionnaire that was promoted on social networking services (SNSs) and the Internet in general. The average age of the participants was 47.77 ± 4.97 years.

The participants' characteristics are presented in Table 1. Among the participants, 195 were in their 40s (68.4%) and 85 were over the age of 50 (31.6%). Their marital status was as follows: 16 unmarried (5.6%), 240 married (84.2%), 5 widowed (1.8%), 21 divorced (7.4%), and 3 cohabiting (1.1%). A total of 162 (56.8%) participants were Protestant, 25 (8.8%)

were Catholic, 52 (18.2%) were Buddhist, 7 (2.5%) reported believing in other religions, and 39 (13.7%) responded that they were not religious. Among the respondents, 186 (65.3%) were college graduates, 43 (15.9%) were graduate school graduates, 53 (18.6%) were high school graduates, 2 (0.7%) were middle school graduates, and 1 (0.3%) was an elementary school graduate.

**Table 1.** Characteristics of participants (*N* = 285).

| Variables | | Frequency | Percent (%) |
|---|---|---|---|
| **Age** | | | |
| | 40~49 | 195 | 68.4 |
| | 50~60 | 90 | 31.6 |
| **Marital status** | | | |
| | single | 16 | 5.6 |
| | married | 240 | 84.2 |
| | widowed | 5 | 1.8 |
| | divorced | 21 | 7.4 |
| | cohabiting | 3 | 1.1 |
| **Religion** | | | |
| | none | 39 | 13.7 |
| | Protestant | 162 | 56.8 |
| | Catholic | 25 | 8.8 |
| | Buddhist | 52 | 18.2 |
| | other | 7 | 2.5 |
| **Educational attainment** | | | |
| | primary school | 1 | 0.3 |
| | middle school | 2 | 0.7 |
| | high school | 53 | 18.6 |
| | college | 186 | 65.3 |
| | graduate school | 43 | 1.1 |

*2.2. Measures*

2.2.1. Religiosity

Participants' religiosity was measured using the Revised Intrinsic/Extrinsic Religious Orientation Scale (I/E-R: Intrinsic/Extrinsic-Revised), which was used in Jun and Suh's study (Gorsuch and McPherson 1989; Jun and Suh 2012). The I/E-R originally consisted of 14 items and 3 factors: intrinsic religious orientation (I: eight items, e.g., "I pray mainly to gain relief and protection."), extrinsic personal (Ep: three items, e.g., "What religion offers me most is comfort in times of trouble and sorrow."), and extrinsic social (Es: three items, e.g., "I go to church mostly to spend time with my friends."). We performed the confirmatory factor analysis (CFA) for the I/E-R three-factor model (I, Es, and Ep), and the model fit was not satisfactory (TLI = 0.846, CFI = 0.876, and RMSEA = 0.111). Thus, we examined the goodness-of-fit after removing items 10 ("Although I am religious, I don't let it affect my daily life.") and 14 ("Although I believe in my religion, many other things are more important in life."); with low SRWs in intrinsic religiosity, it showed the improved fit (TLI = 0.945, CFI = 0.957, and RMSEA = 0.073). Because these two items were classified as extrinsic religiosity in Darvyri et al.'s study (Darvyri et al. 2014), we decided to remove them from intrinsic religiosity. The original scale used a five-point Likert scale, whereas each item in Jun and Suh's study was rated on a seven-point Likert scale ranging from 1 (not at all true) to 7 (very true) to maximize variance (Jun and Suh 2012). In this study, the internal consistencies (Cronbach's $\alpha$) of I, Ep, and Es were 0.88, 0.81, and 0.86, respectively.

2.2.2. Existential Consciousness

Existential consciousness was measured using the existential well-being subscale of Paloutzian and Ellison's Spiritual Well-Being Scale (SWBS), which was used in Jun and Suh's study (Paloutzian and Ellison 1982; Jun and Suh 2012). This scale consists of 20 items

and 2 subfactors: religious well-being (10 items) and existential well-being (10 items, e.g., "Life doesn't have much meaning," and "I believe there is some real purpose for my life."). We selected the existential subscale as existential consciousness because it was divided into a sense of meaningfulness and sense of fullness. Each item was rated on a six-point Likert scale ranging from 1 (not at all true) to 6 (very true). In this study, the internal consistency of the seven items (Cronbach's α) was 0.87.

### 2.2.3. Savoring Beliefs

To measure savoring beliefs, we used the Bryant's Savoring Beliefs Inventory (BSBI: Bryant 2003). We used the translated and validated scale for Koreans developed by Cho et al. (2010). This scale consists of 24 items and 3 sub-factors: anticipation (eight items, e.g., "I can feel the joy of anticipation."), savoring the moment (eight items, e.g., "I know how to make the most of a good time."), and reminiscence (eight items, e.g., "I feel disappointed when I reminisce."). Twelve items, or half of the items in this scale, were reverse scored. In this study, the CFA for the BSBI revealed a poor model fit for the three-factor model. We used only the total score of 20 items, after removing 4 items, which showed low SRWs in the one-factor model. Each item was rated on a seven-point Likert scale ranging from 1 (*not at all true*) to 7 (*very true*). In this study, the internal consistency of these 20 items (Cronbach's α) was 0.93.

### 2.2.4. Subjective Well-Being

Participants' subjective well-being was measured using the Satisfaction with Life Scale (SWLS) developed by Diener et al. (1985) and the Emotion Frequency Test (EFT) developed by Cho and Cha (1998). The SWLS is an instrument designed to measure global cognitive judgments of people's lives. It consists of five items (e.g., "In most ways my life is close to my ideal.") using a 1 (*strongly disagree*) to 7 (*strongly agree*) rating scale. In this study, the internal consistency (Cronbach's α) for these five items was 0.84. The EFT asks participants how frequently they had experienced negative and positive emotions during the last month using ratings from 1 (*not at all*) to 7 (*always*). Participants responded to four types of negative emotions, such as anxiety and depression, and positive emotions, such as joy and emotional intimacy. In this study, the internal consistencies (Cronbach's α) were 0.75 for negative emotions and 0.81 for positive emotions. The level of subjective well-being was determined by subtracting the negative emotion score from the score obtained by adding life satisfaction and positive emotions.

### 2.3. Procedure

Before gathering the data, we obtained approval from the institutional review board (IRB approval number: 2-1040781-A-N-012020043HR), and all procedures were conducted ethically. Data were collected, along with the elements of written informed consent that were presented to the participants, online. Participants were informed that even those who agreed to participate in the online survey could withdraw at any time while answering the questionnaire if they experienced any inconvenience.

### 2.4. Statistical Analysis

The data were analyzed with IBM SPSS Statistics for Windows 26.0, and PROCESS Macro 3.5 was used for this study. The means, standard deviations, skewness, and kurtosis of the data were checked for parametric statistical analyses. CFAs were performed using AMOS. In the CFA, the goodness-of-fit was assessed using the Tucker–Lewis index (TLI), comparative fit index (CFI), and root mean square error of approximation (RMSEA). In general, a RMSEA value of <0.08 suggests a satisfactory model fit. Additionally, a TLI and CFI larger than 0.90 suggests a good model fit (Kline 2005). Pearson's product-moment correlation analysis was conducted using SPSS, and the analysis of a sequential double moderating mediating effect was performed using the PROCESS Macro 3.5 Model 6 (Hayes

2018). Finally, bootstrapping using 5000 resamples with a 95% confidence interval was used to analyze the significance of the mediating model.

## 3. Results

### 3.1. Relationship among Variables Involved in the Subjective Well-Being of Middle-Aged Women

Table 2 presents the correlational analysis of religiosity, existential consciousness, savoring beliefs, and subjective well-being among middle-aged Korean women. None of the absolute values for skewness or kurtosis exceeded 1, indicating that all variable variances were close to the normal distribution needed for conducting parametric statistical analyses (West et al. 1995).

**Table 2.** Correlational matrix of religiosity, existential consciousness, savoring belief, and subjective well-being of middle-aged Korean women ($N$ = 285).

| Variables | 1 | 2 | 3 | 4 | 5 | 6 | 7 | 8 | 9 |
|---|---|---|---|---|---|---|---|---|---|
| 1. | 1 | | | | | | | | |
| 2. | 0.541 *** | 1 | | | | | | | |
| 3. | 0.189 ** | 0.327 *** | 1 | | | | | | |
| 4. | 0.419 *** | 0.144 * | −0.150 * | 1 | | | | | |
| 5 | 0.199 *** | 0.090 | −0.230 *** | 0.683 *** | 1 | | | | |
| 6. | 0.113 | 0.017 | 0.026 | 0.667 *** | 0.606 *** | 1 | | | |
| 7. | 0.214 *** | 0.067 | 0.146 * | 0.576 *** | 0.425 *** | 0.784 *** | 1 | | |
| 8. | 0.221 *** | 0.118 * | −0.014 | 0.587 *** | 0.558 *** | 0.759 *** | 0.503 *** | 1 | |
| 9. | 0.134 * | 0.123 * | 0.050 | −0.382 *** | −0.404 *** | −0.744 *** | −0.362 *** | −0.261 *** | 1 |
| *M* | 28.93 | 14.17 | 8.40 | 50.08 | 104.04 | 19.54 | 16.46 | 20.64 | 17.55 |
| *SD* | 9.31 | 4.46 | 4.67 | 10.84 | 20.47 | 10.93 | 4.23 | 4.77 | 5.36 |
| Skewness | −0.49 | −0.67 | 0.62 | −0.30 | −0.18 | −0.10 | −0.36 | −0.42 | −0.20 |
| Kurtosis | −0.73 | 0.20 | −0.65 | −0.32 | −0.80 | 0.14 | −0.30 | −0.19 | −0.47 |

* $p < 0.05$, ** $p < 0.01$, *** $p < 0.001$. Note: 1 = Intrinsic religiosity, 2 = Extrinsic personal religiosity, 3 = Extrinsic social religiosity, 4 = Existential consciousness, 5 = Savoring beliefs, 6 = Subjective well-being, 7 = Life satisfaction, 8 = Positive emotions, 9 = Negative emotions.

The correlational analysis revealed that intrinsic religiosity was positively correlated with life satisfaction ($r$ = 0.214, $p < 0.001$), positive emotions ($r$ = 0.221, $p < 0.001$), and negative emotions ($r$ = 0.134, $p < 0.05$), whereas extrinsic personal religiosity was positively correlated with positive emotions ($r$ = 0.118, $p < 0.05$) and negative emotions ($r$ = 0.123, $p < 0.05$). Additionally, extrinsic social religiosity was positively correlated only with life satisfaction ($r$ = 0.146, $p < 0.05$).

Intrinsic religiosity ($r$ = 0.419, $p < 0.001$) and extrinsic personal religiosity ($r$ = 0.144, $p < 0.05$) were positively correlated with existential consciousness, whereas extrinsic social religiosity was negatively correlated with it ($r$ = −0.150, $p < 0.05$). Furthermore, intrinsic religiosity was positively correlated with existential consciousness ($r$ = 0.199, $p < 0.001$), whereas extrinsic social religiosity was negatively correlated with it ($r$ = −0.230, $p < 0.001$).

Existential consciousness was closely correlated with subjective well-being ($r$ = 0.667, $p < 0.001$). These two variables accounted for approximately 45% of the total variation. Savoring beliefs were positively correlated with subjective well-being ($r$ = 0.606, $p < 0.001$) and shared approximately 36.7% of the variation.

### 3.2. Verification of the Double Mediation Model for Subjective Well-Being among Middle-Aged Women

This study examined the mediating effect of existential consciousness and savoring beliefs on intrinsic religiosity and subjective well-being among middle-aged Korean women (Table 3), because there were not assumed relationships between extrinsic religiosity and existential consciousness or savoring beliefs in this study. It is known that multicollinearity problems occur when the tolerance is less than 0.2 and the variance inflation factor (VIF) is greater than 5 (O'Brien 2007). Because the tolerance of predictors in this study was

0.450–0.810 and VIFs were 1.234–2.221, multicollinearity was not significant. In addition, the Durbin–Watson value was 2.055, which indicated that there was no autocorrelation detected in the sample, as it was close to 2.

**Table 3.** Double mediating effect of existential consciousness and savoring beliefs on the intrinsic religiosity and subjective well-being of middle-aged Korean women.

| Variables | B | S.E. | t | LLCI | ULCI |
|---|---|---|---|---|---|
| **Mediating variable model (Outcome variable: Existential consciousness)** | | | | | |
| Constant | 35.963 | 1.910 | 18.83 *** | 32.2028 | 39.7223 |
| Intrinsic religiosity | 0.488 | 0.063 | 7.76 *** | 0.3642 | 0.6117 |
| **Mediating variable model (Outcome variable: Savoring belief)** | | | | | |
| Constant | 43.807 | 4.730 | 9.26 *** | 34.4967 | 53.1169 |
| Existential consciousness | 1.373 | 0.090 | 15.31 *** | 1.1965 | 1.5495 |
| Intrinsic religiosity | −0.231 | 0.104 | −2.22 * | −0.4371 | −0.0260 |
| **Dependent variable model (Outcome variable: Subjective well-being)** | | | | | |
| Constant | −17.243 | 2.578 | −6.69 *** | −22.3173 | −12.1686 |
| Existential consciousness | 0.571 | 0.063 | 9.11 *** | 0.4474 | 0.6939 |
| Savoring beliefs | 0.136 | 0.031 | 4.42 *** | 0.0754 | 0.1964 |
| Intrinsic religiosity | −0.205 | 0.054 | −3.77 *** | −0.3119 | −0.0979 |

\* $p < 0.05$, \*\*\* $p < 0.001$. Note: LLCI: lower level for confidence interval; ULCI: upper level for confidence interval.

The results show that intrinsic religiosity positively influenced existential consciousness ($B = 0.488$, $p < 0.001$) and negatively influenced savoring beliefs ($B = −0.231$, $p < 0.05$) and subjective well-being ($B = −0.205$, $p < 0.001$) among middle-aged women in this model. Moreover, existential consciousness positively influenced savoring beliefs ($B = 1.373$, $p < 0.001$) in middle-aged women, and it also significantly or directly influenced subjective well-being ($B = 0.571$, $p < 0.001$). In addition, savoring beliefs positively influenced subjective well-being among middle-aged women ($B = 0.136$, $p < 0.001$).

Figure 2 shows that intrinsic religiosity did not significantly influence subjective well-being among middle-aged women; however, it significantly negatively influenced subjective well-being ($B = −0.175$, $p < 0.001$) when existential consciousness and savoring beliefs were added as mediating variables. This means that intrinsic religiosity negatively influences subjective well-being without its indirect effects through existential consciousness and savoring beliefs among middle-aged women.

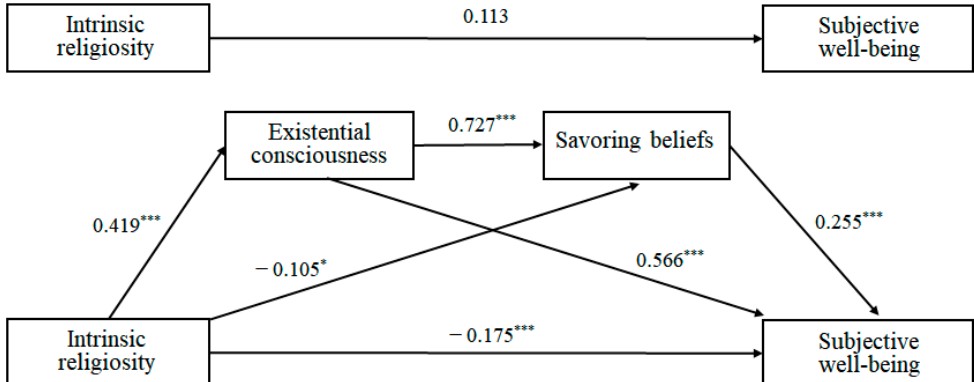

**Figure 2.** The double mediation model of existential consciousness and savoring beliefs on the religiosity and subjective well-being of middle-aged Korean women (\* $p < 0.05$, \*\*\* $p < 0.001$; note: standardized coefficients).

Using 95% bootstrap confidence intervals from 5000 bootstrap replications, the double sequential mediating effect of existential consciousness and savoring beliefs in the relation-

ship between intrinsic religiosity and subjective well-being among middle-aged women was verified, and the results are presented in Table 4.

**Table 4.** Indirect effects of the mediation model.

| Path | Effect | S.E. | BC 95% CI |
|---|---|---|---|
| Total indirect effect | 0.338 | 0.060 | 0.2247~0.4578 |
| Ind1: A → B → D | 0.279 | 0.052 | 0.1816~0.3841 |
| Ind2: A → C → D | −0.032 | 0.163 | −0.0674~−0.0037 |
| Ind3: A → B → C → D | 0.091 | 0.024 | 0.0475~0.1429 |
| Ind1–Ind2 | 0.310 | 0.052 | 0.2138~0.4149 |
| Ind1–Ind3 | 0.187 | 0.057 | 0.0796~0.3042 |
| Ind2–Ind3 | −0.123 | 0.035 | −0.1999~−0.0621 |

Note: A = Intrinsic religiosity, B = Existential consciousness, C = Savoring beliefs, D = Subjective well-being.

In this model, the total mediating effect was 0.338 (0.2247~0.4578), which was evidently significant because no zero existed between the upper and lower bounds of bootstrapping at 95% confidence intervals. Verifying the simple mediating effect revealed that the path from intrinsic religiosity to subjective well-being via existential consciousness was also significant (0.1816–0.3841). The path from religiosity to subjective well-being via savoring beliefs was also significant (−0.0674−−0.0037). Moreover, the sequential double mediating effect of existential consciousness and savoring beliefs on intrinsic religiosity and subjective well-being (intrinsic religiosity → existential consciousness → savoring beliefs → subjective well-being) was 0.091 (0.0475–0.1429), which was significant.

We also analyzed whether there were differences in the indirect effect sizes verified in this study (Table 4). First, the effect sizes of the two indirect paths (A → B → D and A → B → C → D) were larger than that of the second indirect path (A → C → D). In addition, the effect size of existential consciousness on intrinsic religiosity and subjective well-being (A → B → D) was significantly larger than that of the sequential double mediating effect (A → B → C → D) in this study (0.0796–0.3042).

## 4. Discussion

The present study explored the relationships among religiosity, existential consciousness, savoring beliefs, and subjective well-being of middle-aged Korean women. Furthermore, it examined the double mediating effect of existential consciousness and savoring beliefs on the intrinsic religiosity and subjective well-being of middle-aged women. These attempts have produced valuable and useful information for further studies as well as for professionals who help middle-aged women maintain their mental health and promote their quality of life; the implications are discussed below.

First, in this study, the intrinsic religiosity of middle-aged women was not significantly correlated with their subjective well-being. Intrinsic religiosity shared only about 1.3% of the variance ($r = 0.113$) with subjective well-being. This is because among the factors of subjective well-being, intrinsic religiosity had a significant positive relationship with negative emotions as well as subjective well-being and positive emotions. This relationship is evident in the intrinsic religiosity of middle-aged women. It is also noteworthy that extrinsic personal religiosity was positively correlated with both positive and negative emotions, but not with their life satisfaction, whereas extrinsic social religiosity was only correlated with their life satisfaction. For this reason, religiosity may be less closely correlated with subjective well-being than expected in previous studies (Witter et al. 1984). Intrinsic religiosity had less variance shared with negative emotions (1.8%) than with positive emotions (4.9%); however, that intrinsic religiosity had a positive correlation with negative emotions as well as positive emotions among middle-aged women supports the argument that religion is a "double-edged sword."

As hypothesized in this study, middle-aged women's religiosity was positively correlated with both existential consciousness and savoring beliefs. However, in the mediating

model, the path between intrinsic religiosity and savoring beliefs was negatively significant. This means that when adjusted by existential consciousness, the relationship between intrinsic religiosity and savoring beliefs became negatively significant. In other words, the intrinsic religiosity of middle-aged women could not make them experience savoring beliefs without allowing them to experience existential consciousness. These results suggest that existential consciousness plays an important role in the relationship between intrinsic religiosity and savoring beliefs.

However, the positive relationship between religiosity and existential consciousness or savoring beliefs of middle-aged women was evident only in intrinsic religiosity. It was found that extrinsic social religiosity in middle-aged women was even negatively correlated with both existential consciousness and savoring beliefs. This result suggests that middle-aged women's religious orientation toward social interaction and participation in the social network may interfere with the acquisition of existential consciousness and savoring beliefs. A recent study also found that unlike intrinsic religiosity, extrinsic social religiosity did not influence adolescents' well-being through meaning of life (Li and Liu 2021). Wnuk also emphasized only the positive effects of intrinsic and extrinsic personal religiosity, because extrinsic social religiosity was not correlated with hope, prayer, and positive religious coping in his study (Wnuk 2017). Therefore, future studies should focus on the positive aspects of intrinsic religiosity and explore the identity of extrinsic social religiosity in depth.

In this study, the existential consciousness of middle-aged women was closely correlated with their savoring beliefs. Existential consciousness shared approximately 46.6% of the variance ($r = 0.683$) with savoring beliefs. The introduction of savoring as a psychological concept is relatively recent, and research on the influence of savoring beliefs on human quality of life, such as well-being and happiness, has been the primary focus (Burt and Gonzalez 2021; Bryant et al. 2005; Jose et al. 2012). The results of this study highlight the role of existential consciousness as an important factor that can promote savoring beliefs.

We found that savoring beliefs were also closely correlated with subjective well-being in middle-aged women. The results showed that the sequential mediating path from intrinsic religiosity to subjective well-being through existential consciousness and savoring beliefs was significant. However, in this model, the accountability of the direct path from existential consciousness to subjective well-being was large; thus, the influence of savoring beliefs was relatively small. As a result of comparing the indirect effect sizes, the indirect effect size when mediated only by existential consciousness was much greater than the indirect effect size for existential consciousness and savoring beliefs sequentially mediating the intrinsic religiosity and subjective well-being of middle-aged women. This result reiterates that existential consciousness is a determinant of subjective well-being.

A notable finding of this study was that the correlation coefficient between intrinsic religiosity and subjective well-being in middle-aged women was 0.113, which was not significant. However, in the sequential mediating model, intrinsic religiosity negatively affected subjective well-being. These results imply that if intrinsic religiosity is not linked to existential consciousness, it can negatively affect their subjective well-being. Similar results have been reported previously. In Hwang et al.'s study, religious well-being was positively correlated with the psychological well-being of college students, but in a regression analysis with existential well-being added, it had a negative effect on psychological well-being (Hwang et al. 2011). This means that excluding the existential effect of religious well-being negatively affects psychological well-being. In Jun and Suh's study, when religious well-being was included as a predictor of a sense of fullness, it negatively affected people's life satisfaction (Jun and Suh 2012). This suggests that excluding the existential effect of religiosity can negatively affect the quality of human life. This study provides some answers as to what constitutes the edge of the blade that protects people and what constitutes the edge that threatens them in the "double-edged sword of religion". This may be the difference between religiosity with and without existential consciousness.



## 5. Conclusions and Limitations

There are several limitations of this study that should be considered when interpreting the results. First, in this study's sample, the middle-aged women who responded to our online survey could not be considered representative of all middle-aged women. Therefore, it is necessary to confirm the results of this study by sampling other middle-aged women. Furthermore, it is necessary to study the relationships among the variables examined in this study in men and across different age groups. Second, the factorial structures of I/E-R and BSBI were less satisfactory than those revealed in the original scale validation studies. Finally, although we assumed a cause-and-effect relationship between variables based on previous studies, such a relationship cannot be concluded with certainty without the experimental study results. Despite these limitations, this study contributes academically to the literature and clinically to planning interventions to promote the subjective well-being of middle-aged women.

This study found a determinant role of existential consciousness in the relationship between intrinsic religiosity and subjective well-being, as well as between intrinsic religiosity and savoring beliefs in middle-aged Korean women. These results suggest that existential consciousness is a key element of intrinsic religiosity and an important factor in helping middle-aged women develop savoring beliefs to improve their subjective well-being. In addition, extrinsic social religiosity may interfere with middle-aged women by acquiring existential consciousness and savoring beliefs. Thus, mental health professionals should help middle-aged Korean women promote their existential consciousness and avoid social religiosity. In the future, it will be necessary to explore variables, such as some personal traits and cultural factors that can affect existential consciousness.

**Author Contributions:** Conceptualization, E.B. and K.-H.S.; methodology, E.B. and K.-H.S.; validation, K.-H.S. and S.-J.C.; formal analysis, E.B. and K.-H.S.; investigation, E.B.; data curation, K.-H.S.; writing—original draft preparation, E.B. and K.-H.S.; writing—review and editing, S.-J.C. and K.-H.S.; visualization, S.-J.C.; supervision, K.-H.S. and S.-J.C.; project administration, E.B. All authors have read and agreed to the published version of the manuscript.

**Funding:** This research received no external funding.

**Institutional Review Board Statement:** This study was conducted in accordance with the ethical guidelines of the Declaration of Helsinki and was approved by the Institutional Review Board of Sahmyook University (protocol code: 2-1040781-A-N-012020043HR).

**Informed Consent Statement:** All data were collected online with informed consent. The IRB (reference number: 2-1040781-A-N-01) waived the requirement for a documentation signature for the online survey.

**Data Availability Statement:** The datasets analyzed in this study are available from the corresponding author upon reasonable request.

**Conflicts of Interest:** The authors declare no potential conflict of interest concerning the research, authorship and/or publication of this article.

## Abbreviations

| | |
|---|---|
| *SNS* | social networking service |
| *SWBS* | Spiritual Well-Being Scale |
| *SWLS* | Satisfaction with Life Scale |
| *I/E-R* | Intrinsic/Extrinsic Religious Orientation Scale |
| *I* | intrinsic religiosity |
| *Ep* | extrinsic personal religiosity |
| *Es* | extrinsic social religiosity |
| *BSBI* | Bryant's Savoring Beliefs Inventory |
| *EFT* | Emotion Frequency Test |
| *IRB* | institutional review board |
| *SPSS* | Statistical Package for Social Sciences |

| | |
|---|---|
| *AMOS* | Analysis of Moment Structure |
| *TLI* | Tucker–Lewis index |
| *CFI* | comparative fit index |
| *RMSEA* | root mean square error of approximation |
| *VIF* | variance inflation factor |

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
