# Peer review of "Relationship between Religiosity and Subjective Well-Being among Middle-Aged Korean Women: Focused on Roles of Existential Consciousness and Savoring Beliefs"

_religions, doi:10.3390/rel13050402_

Round 1
Reviewer 1 Report
Great research and manuscript! There are a few places were words need to be changed (e.g., research does not "prove" anything, but rather, provides evidence). Also, be sure that you do not imply causation in your results (e.g., line 319). Furthermore, I would encourage you to edit your abstract; it is dense and hard to comprehend. Also, breaking your introduction into sub-headings that align with your variables would be helpful. Finally, I would like to see some clear implications. What do we do with this information now that we know it? Who should do what with it? I would end the manuscript with that information rather than with a discussion about limitations.
Author Response
Reviewer Comments and Responses
Thank you very much for your comments to improve the quality of this articles.
The revised parts were marked in red, and we included the page and line of the revised part.
Response to Reviewer 1 Comments
Point 1. Here are a few places were words need to be changed (e.g., research does not "prove" anything, but rather, provides evidence).
Response 1: Thank you for your great advice. As you advised, we revised the definitive expressions such as 'prove' and 'essential'. (Line 30, 71)
Earlier studies provided some evidences that having religiosity relieves anxiety and depression [6,7]. These findings suggest that religiosity has a positive effect on quality of life.
Because discovering meaning in life can act as an important factor in well-being [25], we believe that existential consciousness may play an important role in people’s subjective well-being
Point 2. Be sure that you do not imply causation in your results (e.g., line 319).
Response 2: Thank you for your comment. Although the mediating model assumes and analyzes the causal relationship, we avoid to state the causal direction because it could not be concluded.. (Line , 331)
In other words, the intrinsic religiosity of middle-aged women could not make them to have a savoring belief without allowing them to experience existential consciousness.
Point 3. I would encourage you to edit your abstract; it is dense and hard to comprehend..
Response 3: We modified the abstract a little bit to make it easier to understand. We edit the abstract based on the guidelines for writing this journal. If you're asking us add sub-headings for a background, method, results, conclusion, we'll do it. (Line 5-19)
Point 4. Breaking your introduction into sub-headings that align with your variables would be helpful.
Response 4: Thank you for your comment. We divided our introduction into sub-headings as below: (Line 24, 61, 90, 112)
1.1. Religiosity and well-being
1.2. Roles of existential consciousness and savoring beliefs in people’s life
1.3. Subjective well-being of middle-aged Korean women
1.4. Hypothesis of this study
Point 5. Finally, I would like to see some clear implications. What do we do with this information now that we know it? Who should do what with it? I would end the manuscript with that information rather than with a discussion about limitations.
Response 5: Thank As you advised, we revised the conclusion part and added some recommendation as below. (Line 382-403)
Thus, mental health professionals need to help middle-aged Korean women promote their existential consciousness and avoid social religiosity. And in future, it is necessary to explore variables, such as some personal traits and cultural factors that can affect their existential consciousness.

Reviewer 2 Report
Overall I found this an interesting and informative article.
I think it would be helpful to have more exploration of and definition of terms here - or make it clearer how you are using them eg religiosity, how does religion overlap with /differ from spirituality/transcendence, existential consciousness.
Be helpful to define existential consciousness more fully here: do you mean having a sense of meaning in life? or a sense of self-awareness/awareness of values?
Need to be careful with assumptions eg line 66 existential consciousness is essential for people's subjective well-being feels like a leap from seeing meaning in life as a factor in wellbeing. Some might argue that consciousness works against religiosity and wellbeing.
on the other hand, it is helpful that you name your assumptions: eg that infusing existential consciousness with savoring beliefs increases subject well-being.
Lines 89 - 93. I assume this is your views? Need to make clear these are assumptions - it could be that one intensely positive experience outweighs many negative ones or that a particular life views carries people through negative life experiences and emotions. It is a big assumption that life satisfaction is more cognitively than emotionally driven. Need to either support with evidence or make clear that that is an assumption you are making.
Lines 90 - 93 Need to make it clear here whether you are referring to what is from other studies or your opinion - i'm not sure you can argue this, it could be that one significantly negative emotion outweighs many positive ones or vice versa.
lines 94-96 be good to be more explicit about what these problems are.
line 101 -given that is quite critical to your research be good to say a little more re reference 36, how big a study was this?
Line 124: should be over the age of 50 not 90.
Line 136 need to explain the intrinsic/extrinsic religion orientation scale more: could give example of each.
Line 152, be helpful for those not familiar with this scale to include a couple of examples from each sub-factors. Line 158, 'sense of fullness' meaning?
Line 161. Also be good to have some examples here.
LIne 356 instead of saying if intrinsic religiosity does not make people experience existential consciousness, would suggest saying something like if intrinsic religioisity is not linked to existential.....
Discussion and conclusion - would be helpful to tease out what the results might mean more fully - ie how might this help with promoting the subject wellbeing of middle aged women in Korea. Are there any cultural factors a play here that would be good to articulate/
Author Response
Reviewer Comments and Responses
Thank you very much for your comments to improve the quality of this articles.
The revised parts were marked in red, and we included the page and line of the revised part.
Response to Reviewer 2 Comments
Point 1. It would be helpful to have more exploration of and definition of terms here - or make it clearer how you are using them eg religiosity. Be helpful to define existential consciousness more fully here.
Response 1: Thank you for your advice. As you advised, we have included a brief definition of religion and existential consciousness as follows. (Line 32-33, 62-63)
Religiosity, which can be defined as a religious orientation that commits to religious beliefs and activities, may improve an individual's psychological well-being [8];
We chose existential consciousness–that is, the sense of existence refers a present state of recognizing the meaning, purpose, and fullness in one’s life [21].
Point 2. Need to be careful with assumptions eg line 66 existential consciousness is essential for people's subjective well-being feels like a leap from seeing meaning in life as a factor in wellbeing.
Response 2: Thank you for your comment. As you advised, we revised the definitive expressions such as 'prove' and 'essential'. (Line 30, 71)
Earlier studies provided some evidences that having religiosity relieves anxiety and depression [6,7]. These findings suggest that religiosity has a positive effect on quality of life.
Because discovering meaning in life can act as an important factor in well-being [25], we believe that existential consciousness may play an important role in people’s subjective well-being
Point 3. It is helpful that you name your assumptions: eg that infusing existential consciousness with savoring beliefs increases subject well-being.
Response 3: Thank you very much for your good comment on this part. The purpose of this study was to verify the model of the three factors. And it was impossible that researchers could choose the number of factors because it was validated and standardized scales with copyright in Multi-Health Systems. Even so, we tried a confirmatory factor analysis with one factor, and this model showed poor model fit. We added these results to Table 2 and this section as follows: (Line 77-78)
Thus, we assumed that the savoring beliefs created by existential consciousness increases subjective well-being.
Point 4. Lines 89 - 93. I assume this is your views? Need to make clear these are assumptions - it could be that one intensely positive experience outweighs many negative ones or that a particular life views carries people through negative life experiences and emotions. It is a big assumption that life satisfaction is more cognitively than emotionally driven. Need to either support with evidence or make clear that that is an assumption you are making.
Response 4: I'm sorry. We didn't suggest a reference by mistake. This statement is based on the assumption of Diener and his colleagues that establishes the concept of subjective well-being and life satisfaction. So, we added the following reference as below: (Line 446-447)
Diener, E.; Emmons, R.A.; Larsen, R.J.; Griffin, S. The satisfaction with life scale. J. Pers. Assess. 1985, 49, 71-75. doi:10.1207/s15327752jpa4901_13
Point 5. Lines 90 - 93 Need to make it clear here whether you are referring to what is from other studies or your opinion - I'm not sure you can argue this, it could be that one significantly negative emotion outweighs many positive ones or vice versa.
Response 5: This part is a more specific restatement of what is described just above. Therefore, the reference is [31].
Point 6. Lines 94-96 be good to be more explicit about what these problems are.
Response 6: That part you commented is already described below. (Line 211-213)
In addition, middle-aged women in Korea experience extensive life-related stress as well as fear of loss and loneliness because they are bound by their families as wives, mothers, housewives, and daughters-in-law [35].
Point 7. Line 101 -given that is quite critical to your research be good to say a little more re reference 36, how big a study was this?
Response 7: As you advised, we restated the sentence in more detail as below. (Line108-109)
It was found that middle-aged women’s sense of existence, such as discovering the meaning of life, was positively correlated with their well-being [38].
Point 8. Line 124: should be over the age of 50 not 90.
Response 8: Shamefully, it was a typo. It has been corrected. (Line 131)
Point 9. Line 136 need to explain the intrinsic/extrinsic religion orientation scale more: could give example of each.
Response 9: Thank you for your valuable comment. As you advised, we have included an example of the items as follows. (Line 147-149)
I/E-R originally consists of 14 items and 3 factors: intrinsic religious orientation (I: eight items, e.g. “I pray mainly to gain relief and protection.”), extrinsic personal (Ep: three items, e.g. “What religion offers me most is comfort in times of trouble and sorrow.”), and extrinsic social (Es: three items, e.g. “I go to church mostly to spend time with my friends.”).
Point 10. Line 152, be helpful for those not familiar with this scale to include a couple of examples from each sub-factors. Line 158, 'sense of fullness' meaning?.
Response 10: As you advised, we have included an example of the items as follows. (Line 166-167)
This scale consists of 20 items and 2 sub-factors: religious well-being (10 items) and existential well-being (10 items, e.g. “Life doesn’t have much meaning,” “I believe there is some real purpose for my life.”).
Point 11. Line 161. Also be good to have some examples here.
Response 11: As you advised, we have included an example of the items as follows. (Line 174-176, 187-188, 192-193)
This scale consists of 24 items and 3 sub-factors: anticipation (eight items, e.g. “I can feel the joy of anticipation.”), savoring the moment (eight items, e.g. “I know how to make the most of good time.”), and reminiscence (eight items, e.g. “I feel disappointed when reminisce.”).
It consists of five items (e.g. “In most ways my life is close to my ideal.”) that use a 1 (strongly disagree) to 7 (strongly agree) rating scale.
Participants responded regarding four types of negative emotions, such as anxiety and depression; and positive emotions, such as joy and emotional intimacy.
Point 12. LIne 356 instead of saying if intrinsic religiosity does not make people experience existential consciousness, would suggest saying something like if intrinsic religiosity is not linked to existential.
Response 12: Thank you for your good suggestion. We modified as you advised as below: (Line 368-369)
These results imply that if intrinsic religiosity is not linked to existential consciousness, it can negatively affect their subjective well-being. Similar results have been reported previously
Point 13. Discussion and conclusion - Are there any cultural factors a play here that would be good to articulate.
Response 13: Thank As you advised, we revised the conclusion part and added some recommendation for future studies, including cultural factors are to be explored, as below. (Line 382-403)
Thus, mental health professionals need to help middle-aged Korean women promote their existential consciousness and avoid social religiosity. And in future, it is necessary to explore variables, such as some personal traits and cultural factors that can affect their existential consciousness.

This manuscript is a resubmission of an earlier submission. The following is a list of the peer review reports and author responses from that submission.